# Coral spawning patterns in the Gulf of Thailand reveal synchronised annual daytime spawning, with a review of spawning patterns in *Pavona* corals across the Indo-Pacific

Rahul Mehrotra[1], Suppakarn Jandang[2,3]*, Coline Monchanin[1], Matthias Desmolles[1], Lalita Putchim[4], Natchanon Kiatkajornphan[1], Supatcha Japakang[1], Anne Groenevelde[5], Morokot Long[6], Matthew Glue[6], Anchalee Chankong[3], Nitchanan Nilkerd[3], Laddawan Sangsawang[3], Vincent Pardieu[1,7]

**1** Aow Thai Marine Ecology Center, Love Wildlife Foundation, FREC Bangkok, Bangkok, Thailand, **2** Research Institute for Applied Mechanics, Kyushu University, Kasuga, Fukuoka, Japan, **3** Center for Ocean Plastic Studies, Research Institute for Applied Mechanics, Kyushu University, Chulalongkorn University Research Building, Pathumwan, Bangkok, Thailand, **4** Marine and Coastal Resources Research Center, Rayong, Thailand, **5** Bubbles Up Dive Center, Preah Sihanouk, Cambodia, **6** Fauna & Flora, Cambodia, Phnom Penh, Cambodia, **7** VP Consulting WLL, Manana, Kingdom of Bahrain

* suppakarn.j@riam.kyushu-u.ac.jp

## Abstract

Global documentation of coral spawning has been increasing, yet remains restricted to corals and reefs for which established patterns have largely been identified, being overwhelmingly associated with periods of darkness after sunset. With this increase in geographic representation has come a growing number of spawning observations that skew or directly challenge previously established trends. After multiple incidental observations of daytime spawning of *Pavona* corals in the Gulf of Thailand, we aimed to establish spawning patterns for *Pavona* in the region using non-invasive surveys. Here we document synchronous daytime coral spawning for the species *Pavona explanulata* and *P. varians* across over 450 km of coastline along the Eastern Gulf of Thailand and Cambodia. Data collected via observation-only SCUBA surveys over a five-year period identified contrasting patterns of spawning timing of these corals from most other taxa for which data is available in the Gulf of Thailand, based on time of the year, proximity to the lunar peak and hours after sunrise. Based on a null-dataset of reef surveys with an absence of spawning, we were able to add credence to a restricted spawning period of these corals to being typically between September and December in the Gulf, with a sequential spawning of *P. explanulata* after *P. varians* with minimal overlap. We identified a consistently predictable onset of gamete release approximately between 6–8 hours after sunrise for *P. varians* and 7–9.5 hours for *P. explanulata*. We therefore conducted a review of spawning timing and patterns for *Pavona* corals across the globe identifying considerable variation across taxa and regions with support for high synchronicity within biogeographic regions.

**Data availability statement:** Data for this research is included as supplementary files. Supplementary video files are available on Figshare at https://doi.org/10.6084/m9.figshare.29945111 and https://doi.org/10.6084/m9.figshare.29711513.

**Funding:** This research was supported by Love Wildlife Foundation, Thailand and Fauna and Flora International, Cambodia.

**Competing interests:** The authors have no competing interests to declare that are relevant to the content of this article.

Finally, our dataset also incidentally documented additional cases of synchronised daytime gamete release from *Cycloseris* and *Porites* corals highlighting the need for further study on the dynamics of daytime gamete release.

## Introduction

Scleractinian corals reproduce asexually and sexually. Asexual reproduction occurs due to fragmentation and results in colony formation and growth. Sexual reproduction contributes to population turnover and adaptation through natural selection, and happens either externally (broadcasting) or internally (brooding). Sexual reproduction by both external and internal means rely on the release of sperm and hydrodynamics to disperse gametes across the reef. It is estimated that the majority of known scleractinian corals undergo simultaneous hermaphrodite broadcast spawning, releasing both eggs and sperm into the water column for external fertilization [1–3]. Usually, this involves mass spawning events, which is thought to promote successful fertilization [4] and increased survival due to over-saturation of prey [5]. Mass spawning events can involve dozens of species of corals in a given reef in a single night, with over 100 species spawning in a spawning season [6]. Furthermore, most broadcasting coral species spawn once a year as opposed to the two (or more) spawning peaks seen in some corals [3,7]. These events require synchrony across multiple colonies of a given species to be successful, however a wide array of conditions and cues have been linked to the synchrony of such events [8–13], most predominantly the role of temperature [14,15] and the lunar cycle [6,16].

Solar insolation cycles have been considered a key factor in determining spawning synchrony for both month of the year and time of the day [9,17], meanwhile the lunar cycle is thought to be a major influence in synchrony for date (typically in the form of days after full moon/new moon) [2,18,19]. The precise mechanics of how the lunar cycle mediates or influences development and spawning in corals remains a topic in need of further exploration despite irrefutable evidence of widespread association between full or new moon conditions and spawning timing [20,21]. Much attention has been paid to the role of illumination (or lack of) as the factor in cueing spawning timing and the disruptive effects of artificial illumination on localised spawning [21–23]. Recent work has shown that the absence of light (i.e., periods of darkness after sunset) as opposed to illumination by the full moon may act as a primary cue for the onset of spawning of certain corals [20,24]. Further support for lunar illumination contributing to synchrony comes from evidence of increased expression of genes associated with photoreception during full moon compared to new moon nights [22,25,26].

While investigations into reproductive cues and environmental effects on spawning have continued to increase in complexity, so too have we increasingly seen examples of coral reproduction that broaden conventionally accepted spawning periods. Growing reports on corals spawning during daylight or twilight conditions challenge the understanding in the role of illumination cycles [21,27–30]. The most in-depth and intensive exploration of daytime coral spawning synchrony was recently documented over a decade-long dataset of *Porites rus* (Forskål, 1775) revealing large-scale

synchrony in coral spawning in the southern hemisphere [21]. Among the earliest examples of synchronised daytime coral spawning in-situ documented multiple colonies of *Pavona gigantea* (Verrill, 1869) spawning 10 hours after sunrise from the Galapagos islands [31], and colonies of '*Pavona* sp.' spawning shortly before midday in the Gulf of Thailand [30]. While an earlier documentation of *Pavona varians* (Verrill, 1864) spawning shortly before sunset was made from Hawaiian waters [32], no other cases of explicitly daytime spawning has been recorded in the literature among the Agaraciidae, let alone the genus *Pavona* Lamarck, 1801. The reproductive dynamics of most described scleractinian corals remains unknown, and biogeographical variation for only a few taxa have been explored [21,33]. Additionally, ongoing reshuffling of scleractinian taxonomy and systematics as part of the 'molecular revolution' [34,35] have revealed multiple species complexes, often with significant implications on their reproduction [36–38]. Indeed the recently describe *Pavona giannii* Benzoni, 2025 has historically been confused with *P. varians* [39], highlighting ongoing taxonomic progress within the genus.

Few published records exist regarding data on mass broadcast coral spawning in Thailand [40–42]. Recent multi-year surveys at the island of Koh Tao have relied on spawning periodicity and demonstrated the use of observational baseline data to predict annual spawning events for specific coral genera, to within 30–60 minutes or less, using SCUBA diving [43]: Appendix A). The most comprehensive dataset for coral spawning, both locally for Thailand and globally, was recently compiled by Baird et al [33]. Most earlier work from Thai waters have relied on taxon-specific or genus-specific assessments of gamete development to determine spawning period which is a common practice among reef research and management groups in Thailand.

The standard protocol for monitoring of reproductive development and spawning among researchers and managers in Thailand remains to be the utilisation in-situ histology, or careful breaking of mature colonies of specific genera, to visualise gamete bundles within colonies [40–42]. This long-held practice is still used around the world today to quantify and qualify reproductive development, and to predict coral spawning events [23,44,45]. While this technique provides a high-precision method to assess presence/absence and development of gametes in many corals, it remains an intrusive practice and likely a stressor to parent colonies during spawning periods. A potentially less destructive technique has been carried out at the island of Koh Mun Nai, in the Gulf of Thailand, involving the temporary relocation of small mature colonies from in-situ to ex-situ conditions to observe gamete release (and maximise gamete collection), before being returned to the reef ideally undamaged [46]. This, however, often requires the same physical damage to a colony to verify gamete development.

In the present study, we investigated spawning patterns for corals across the Eastern Gulf of Thailand using entirely non-invasive methods and identified distinct daytime spawning and night-time spawning periods across different groups. Our data contributes to the growing observations of daytime spawning in scleractinian corals, and in contextualised by a diversity of night-time spawning corals from the same sites. In particular, the documentation of daytime spawning of *Pavona explanulata* (Lamarck, 1816), *Pavona varians*, *Cycloseris* sp. and *Porites* sp. (likely *P. lobata*). Based on our findings, we aimed to then delineate the spawning periods of *Pavona* corals in the region using only predictive and in-situ observational techniques (as opposed to periodic histological sampling of wild colonies). Finally, we compared these findings from our five-year dataset to the global available data on *Pavona* coral spawning timing and discuss the potential significance of daytime spawning records from the region.

## Materials and methods

### Establishing a baseline

To develop predictive power of the spawning time for the greatest diversity of scleractinian corals, a baseline of optimal spawning window was established for scleractinian corals in the Gulf of Thailand. Using a combination of our own unpublished data and previously published records on gamete development and broadcast spawning observations from the Gulf of Thailand [30,40,43,46], distinct spawning windows were identified. This was supported by observations by the

monitoring teams of the Department of Marine and Coastal Resources (DMCR, pers. Comm.; S1 Table). The resulting dataset identified two discrete spawning periods during the year and revealed that the spawning and development information for the Agariciidae was represented by only two species of *Pavona* corals, both spawning during daytime. Using this baseline, a more specific investigation was carried out to 1) identify the specific spawning time for *Pavona* corals along the Eastern Gulf of Thailand and 2) investigate other instances of daytime spawning within the Gulf of Thailand.

## Defining spawning periods

To test the reproducibility and define general spawning window of corals, in-situ roving surveys were carried out using SCUBA at 30 islands across three provinces (Chonburi, Rayong and Trat) along the Eastern Gulf of Thailand between 2021 and 2024. It is important to note that surveys were restricted to periods between January and April, and September and December, as annual monsoon conditions between May and August prevented access to sites and SCUBA surveys. These surveys were carried out at fringing coral reefs and pinnacles with the aim of observing coral spawning (gamete release), with data on time of spawning, gamete type (sperm, eggs, bundles), species (where readily identifiable), and other metadata collected (S1 Table). The majority of surveys were conducted during the daytime to identify daytime spawning corals, with night-time surveys typically carried out during established spawning periods. Night-time surveys in the months of March and April 2023 and 2024 were heavily supplemented by the use of a remotely operated vehicle (ROV) due to growing observations and incidents of *Chironex* sp. box jellyfish (S2 Fig) [47,48]. Spawning corals observed via ROV were initially identified to genus and tabulated during the survey, and then were later verified or disregarded based on review of the footage (S3 Fig), with the exception of a single survey where footage recording was corrupted.

Additionally, surveys with multiple parallel data collection efforts such as reef monitoring [e.g., 49,50], discarded fishing gear removal [51] etc., which were often being carried out concurrently at these same sites, are included in our dataset. These surveys, while often not focused on observing coral spawning events, did nonetheless find presence and absence of daytime spawning and are thus included in the dataset with the assumption that mass synchronised spawning of specific taxa have the potential to be detected by trained surveyors (and incidentally by recreational divers). We have thus referred to any survey in this broader dataset, or any survey explicitly looking for spawning but where spawning was not observed, as 'null data'. Null data surveys should be considered as surveys where multiple colonies were directly observed, typically briefly and usually as transect or roving-diver surveys, where no gamete release was observed. Null data surveys also included more prolonged observations of individual colonies where an absence of spawning was noted. Therefore, null data observations may be considered as defined and specified periods of time where multiple colonies across multiple taxa were specifically assessed for varying periods of time without any indication of sexual reproductive activity. The baseline community structure of corals for all null data survey sites was recently assessed to contextualise reef composition [52]. In addition, four incidental observations of daytime coral spawning from the Gulf of Thailand by academically inclined members of the recreational diving community were submitted to the authors for review and subsequent inclusion in the dataset. No observations were included where photo/video evidence, precise timing of spawning, and number of spawning colonies were lacking.

Corals were identified to species level where possible. However, night-time spawning corals are presented only at the genus level in this study, owing to the focus on daytime observations and the difficulty of accurate taxonomic identification without sampling. Identification was conducted during daytime surveys across the reefs, based on living specimens or, where possible, the skeletal morphology of recently killed corals (S4 Fig), using original descriptions and relevant literature (e.g., [34,38,53,54]) for corals. Taxonomy was updated according to World Register of Marine Species (WoRMS) [55]. Daytime spawning corals were identified to species level through close in-situ examination of skeletons and colony morphology. Gravid colony assessments and validation were not performed due to the rapidly declining coral populations at the survey sites [51,56,57].

### *Pavona* surveys

After the first year of surveys, general patterns of spawning were identified for the two species of *Pavona*, *P. explanulata* and *P. varians*. Therefore, alongside aforementioned generalised day and night-time surveys, dedicated surveys between 2022–2024 were carried out to study the spawning of *P. explanulata* and *P. varians* across the Eastern Gulf of Thailand, including two islands in Cambodian waters. These surveys were carried out in October, November and December, between 3 days before the lunar peak and 3 days after the lunar peak, between 12:00 and 17:00, (based on weather conditions). The purpose of these surveys was to investigate the synchrony of gamete release across the > 450km of coastline along the eastern Gulf of Thailand, encompassing both Thailand and Cambodian waters, and to develop a more robust understanding of reproductive outputs of *Pavona* corals in the region.

### Data analysis

To review and compare findings from our surveys and explore biogeographical or taxonomic trends, spawning data were collected from all available published literature globally on *Pavona* spawning [31,36,43,58–62]. For all data including spawning observations from present study and previous studies compiled, and null data, the following information was collected where relevant: location; identity of spawning coral (to the most reliable taxonomic level); precise date and time of observation duration, gamete bundle setting; and release. Where possible, the number of colonies and last observed spawning time were also collected. From these spawning data were then calculated the number of minutes and hours of spawning onset since the most recent sunrise, the number of hours since the nearest full moon (FM), and the number of days since the start of the year (as a higher resolution proxy for month and season). Precise data for sunrise time and FM timing was gathered and cross-referenced from online and published sources (see [33]). It is important to note that data used for this purpose utilised lunar peak timing as defined by syzygy of the Sun-Earth-Moon system as opposed to by illumination. Therefore, we are able to greatly increase the precision of our calculations by using minutes after lunar peak, rather than nights after the nearest visible FM to greatly improve the resolution of spawning predictions. Data were then visualised with plots generated using XLstat 2024.4.1.

### Results

A total of 694 corals/colonies were recorded to be spawning along the Eastern Gulf of Thailand across the five-year survey period (S1 Table). Of these, 366 were recorded spawning during daylight hours (between sunrise and sunset) and 328 colonies were recorded setting or spawning after sunset. Daytime spawning coral observations (Fig 1) were dominated by colonies of *Pavona explanulata* (n = 116) and *P. varians* (n = 232), with additional observations including 13 colonies of *Porites* sp. (submitted through community observations), four individuals of *Cycloseris* sp. and a single colony of '*Dipsastraea*' sp. recorded by the authors (Fig 2G). All observations of *Pavona* spawning occurred during daylight hours (between 7 and 9.5 hours after sunrise), with *P. varians* recorded between 12:08 and 14:44 and *P. explanulata* starting no earlier than 13:12 with the last spawning recorded at 15:44. Overlap of spawning period between the two species of *Pavona* was recorded on only a single day, on the 11th of October 2022, where the last spawning of *P. varians* was recorded at 14:44 and the earliest spawning of *P. explanulata* was recorded at 14:37, resulting in a seven-minute overlap. Additionally, all *Pavona* spawning was recorded between the 30th of September and 27th of December, with the majority of all other spawning records occurring during the 'conventional' spawning period of between February and April. A conspicuous absence of all other Agariciidae corals across the entirety of our dataset was noteworthy. Outliers from these clusters include the four *Cycloseris*, observed between 12:49 and 12:56 (over 6 hours after sunrise) in January, a single *Dipsastraea* spawning at 15:20 alongside *P. explanulata* in October, and a single *Hydnophora* spawning at 19:14 in November. The *Dipsastraea* spawning was classified based on continuous release of sperm (putative) from a single colony, with no signs of gamete bundles. The taxonomic attribution is tentative given the numerous complexities in the in-situ identification of this group [65]. Of significant note are the 13 colonies of *Porites* (likely *P. lobata*) observed at a single site (Sail Rock)

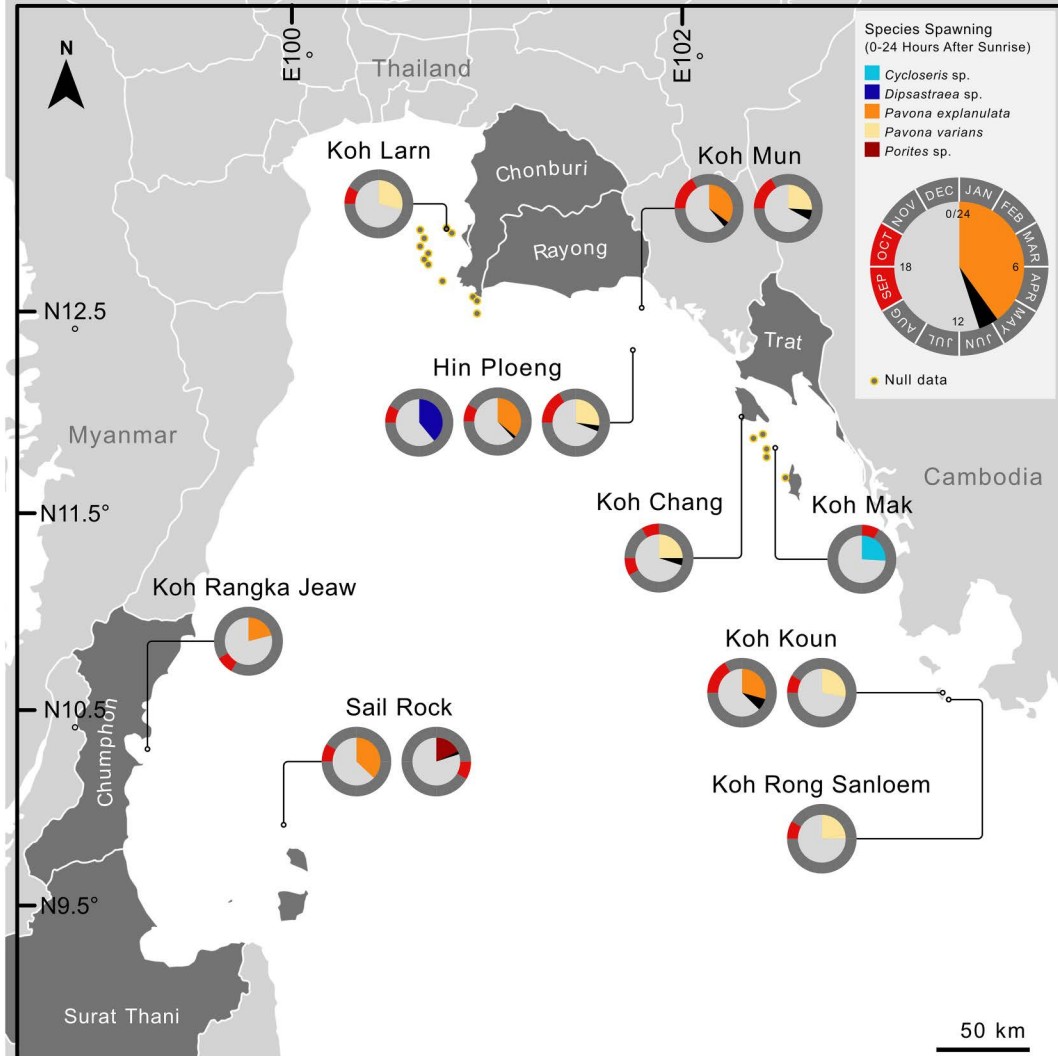

**Fig 1. Daytime spawning corals across the Gulf of Thailand.** Dataset includes present surveys and previous records from the Western Gulf of Thailand. Coloured section of charts indicate briefest quantity of minutes and hours after sunrise (out of a total 24) at the start of spawning, with latest recorded start of spawning for each coral and location indicated in black. Outer ring of the pie chart indicates the total months of the year during which spawning was identified per species per location. Sites which were surveyed but where no spawning was recorded are identified as null data. Legend provides an example of *Pavona explanulata* indicating earliest start of spawning period to be approx. 9 hours and 50 minutes after sunrise, with the latest start of spawning being approx. 10 hours and 50 minutes after sunrise. Legend indicates all spawning records to be in September and October. Pie charts without black sections indicate only a single spawning date recorded or identical spawning onsets. Detailed timings accessible in S1 Table. The basemap incorporates provincial boundary data from SimpleMaps [63,64]. (CC BY 4.0).

spawning six hours after sunrise over two consecutive days in April 2023, whereas the vast majority of *Porites* spawning in the Gulf of Thailand was recorded after sunset.

The majority of daytime spawning observations made in this study were of seemingly gonochoric colonies. Colonies of *Pavona* were typically found to be releasing sperm or eggs exclusively, with between 1–5% of colonies (per event) observed as simultaneous hermaphrodites (Fig 2B). In such cases, eggs and sperm were conspicuously released from different parts of the same colony, often with a defined area in between with no gametes being released. Eggs were

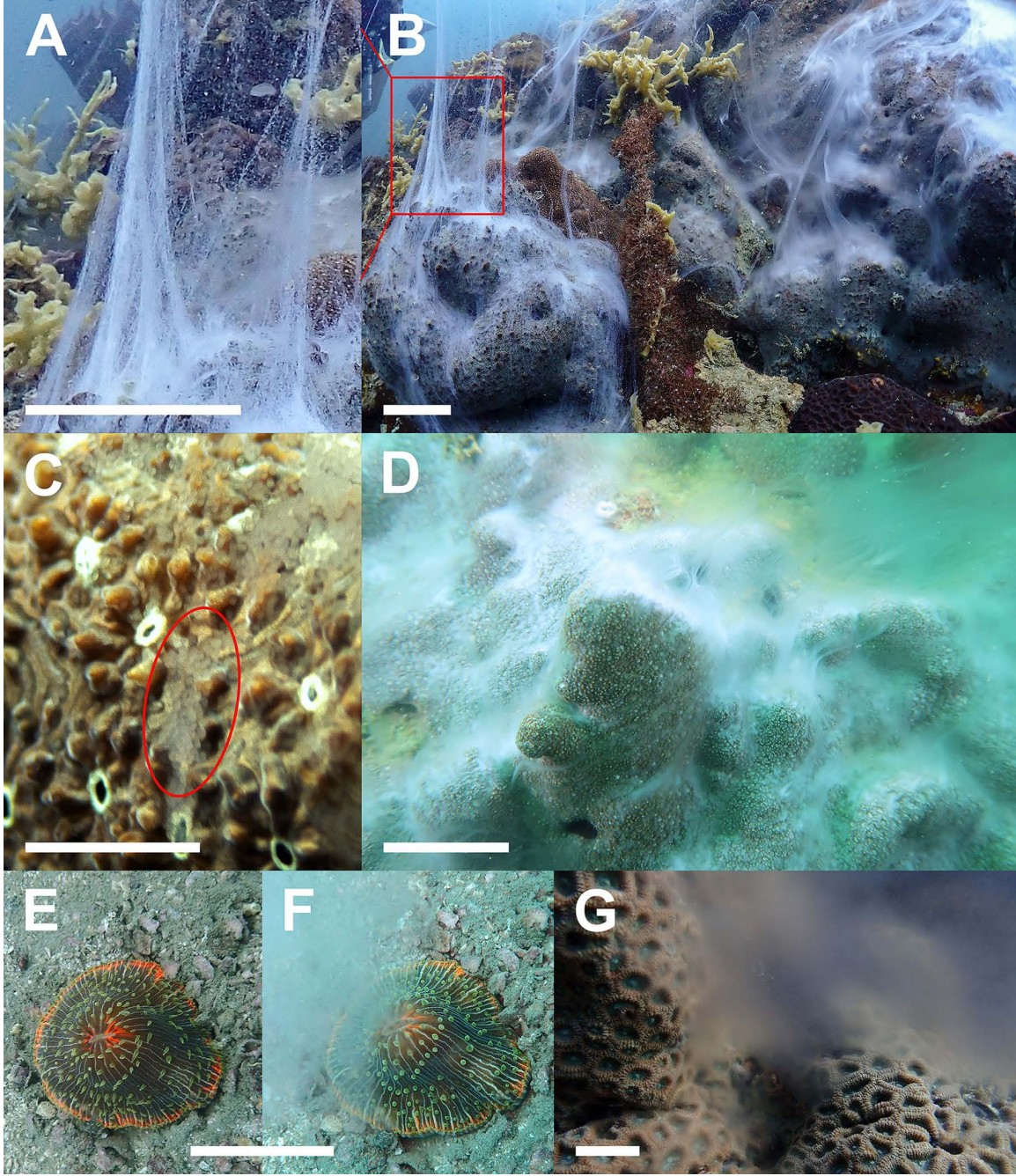

**Fig 2. Photographs of daytime gamete release of corals recorded along the Eastern Gulf of Thailand.** A) Egg release by partial colony of *Pavona varians* (scale bar 50 mm); B) Simultaneous egg and sperm release by a hermaphroditic colony of *P. varians* (scale bar 50 mm), red box indicates portion zoomed in for (A) to more easily differentiate egg release from sperm; C) Near-transparent egg-release (red ellipse for example) by polyps of *P. explanulata* (scale bar 10 mm; photo by Thomas Thana Real); D) Whole-colony sperm release by *P. explanulata* (scale bar 50 mm); E) and F) Photos taken five seconds apart before and during gamete release by *Cycloseris* sp. (scale bar 50 mm); G) Sperm release by *Dipsastraea* sp. (scale bar 50 mm).

released in mucus strings/tendrils, largely in agreement with the description by Glynn et al. [36], with eggs barely visible to naked eye (Fig 2A), initially being easily confused for sperm release. Egg tendrils however did not result in diffuse clouds, rather being dispersed increasingly diminished tendrils before finally diffusing further away from the colony-surface. Among gonochoric colonies, gamete release was typically observed across the majority of the colony's surface, however it was common to observe partial gamete release, particularly in observations made in 2024 after widespread bleaching and mortality recorded in the region [36,66]. In cases where gamete release was partial, some attempts were made to return to the colonies at frequent intervals to observe if one gamete release would precede the other, however no such cases were observed, and these were not carried out with dedicated colony tagging nor fixed protocol. Between the two species, sperm release was equally visible from a distance, however egg release in *P. varians* was observed to be considerably more visible due to the higher density of eggs released in the tendrils and the opaque-white colour of the eggs. In contrast, eggs of *P. explanulata* were far less obvious, with both eggs and tendrils being smaller and often more translucent (Fig 2C) than typically seen in *P. varians*. Finally, it is perhaps noteworthy that all other daytime spawning non-agariciid corals in our surveys were exclusively found to be releasing sperm, with no sign of eggs/bundles.

All daytime spawning corals recorded in our dataset were found to be spawning between 77 hours (3.2 days) before and 71 hours (3 days) after lunar peak, whereas almost all corals found to be spawning during or after sunset (>11.5 hours after sunrise) were recorded between 92 hours (3.8 days) and 162 hours (6.8 days) after lunar peak (Fig 3). A single colony of *Platygyra daedalea* was recorded to spawn at 73 hours (3 days) after lunar peak in 2024, however, this is an anomaly relative to all other *Platygyra* records (n = 104 colonies) which were setting or spawning between 102 and 162 hours after lunar peak. Colonies of *Pavona varians* were observed spawning between 77 hours (3 days and 5 hours) before and 68 hours (4 hours before the fourth day), after lunar peak, and colonies of *P. explanulata* were recorded between 44 hours before and 45 hours (between 3–4 hours before the third day) after lunar peak (Fig 4). No discernible differentiation was observed in the spawning time or period between the northernmost colony (Chonburi province, Thailand) and the southernmost colonies (Koh Rong Sanloem, Cambodia) despite over 450 km of coastline separating the sites.

Our observations reveal no indications of biogeographic clustering within the eastern Gulf of Thailand, with the northernmost colony (Chonburi province, Thailand) spawning at a similar time of day, time of year, and lunar period as 'central' and southernmost observations (Koh Rong Sanloem, Cambodia). Along the western Gulf, the single observation of *P. explanulata* from Koh Tao [43] also largely agrees with this period (though spawning in September), however the seven colonies observed at Chumphon province [30] remain as notable outliers due to spawning earlier in the day, year, and lunar period than all other *Pavona* records from the Gulf (Figs 1,4). However future surveys in the months of May to August (outside of the typical monsoon period for the western Gulf) may reveal trends that expand this period. When compared with all currently recorded *Pavona* spawning around the globe, our findings represent the earliest spawning in the day (hours after sunrise) yet the latest in the year, typically after 270 days after the new year (Fig 5). The majority of global records cluster between 3 days before and four days after FM, with exceptions including observations from Japan and Taiwan both spawning later than 5 days after FM [58,60,61]. Additionally, a single record of *P. varians* from Panama was recorded as 97 hours (4.04 days) after FM [34], and the observation of multiple colonies of *Pavona* sp. [30] (identified here as *P. explanulata*) as an outlier to wider dataset (see Discussion). Furthermore, these Central and East-Pacific observations record *P. varians* spawning in the first half of the year (Jan-May for Panama, June for Hawaii) whereas the entirety of *P. varians* spawning from the Eastern Gulf of Thailand was recorded between September and December. Overall, both taxa *P. explanulata* and *P. varians* were recorded to be spawning across various times of the year and times of the day across the globe, with most observations of *Pavona* corals outside of the Gulf of Thailand occurring at least 12 hours after sunrise (Figs 5, 6). Interestingly, observations of *P. gigantea* from the Galapagos Islands [31] occurred during the daytime (approx. 10 hours after sunrise) and *P. clavus* from Contadora Island in Panama [59] were recorded during sunset (average of 11 hours and 45 minutes after sunrise).

 

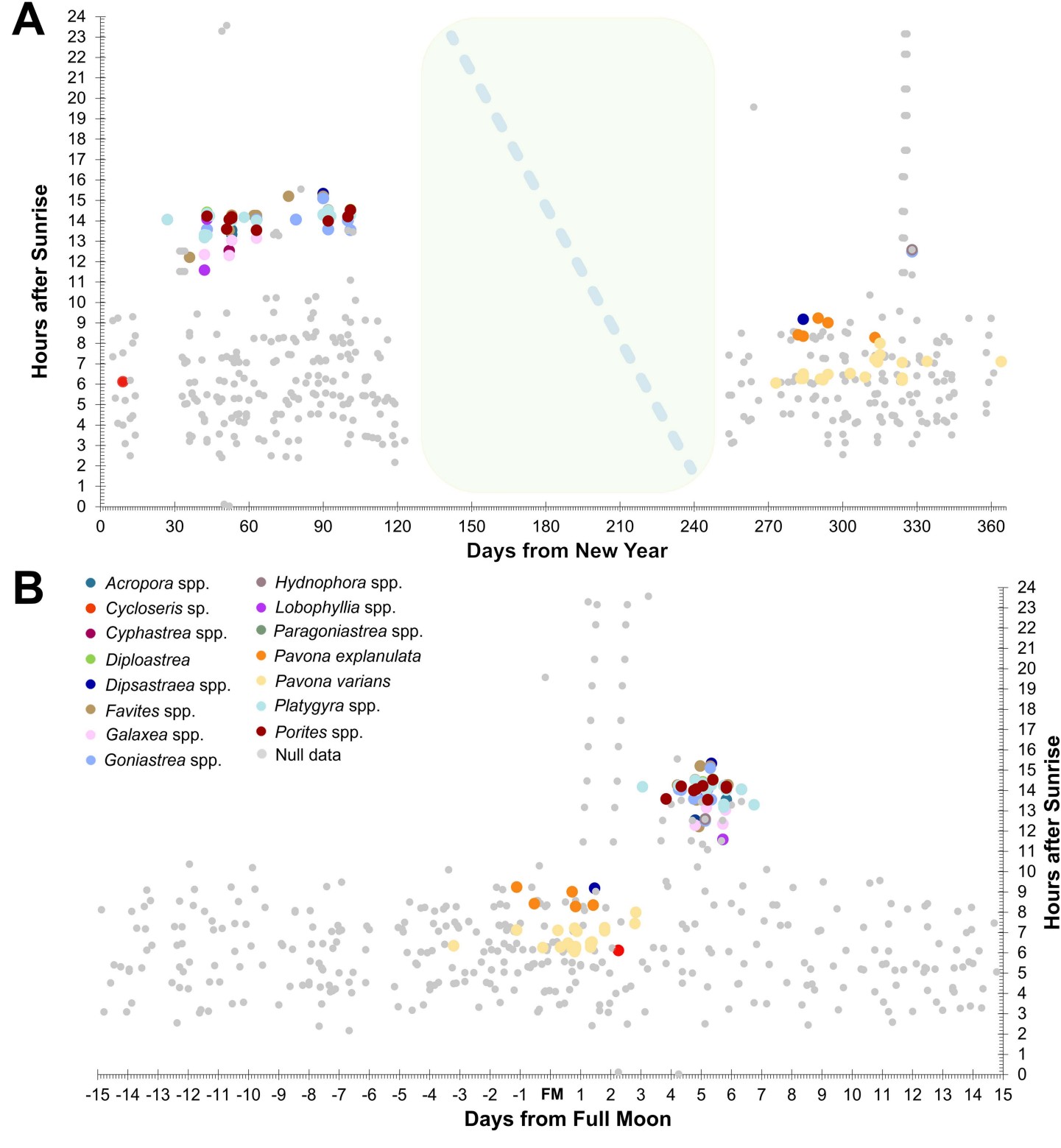

**Fig 3. Plots of spawning timing of all recorded scleractinian corals, and null data, along the Chonburi, Rayong and Trat provinces of the Gulf of Thailand between 2021 and 2024.** A) Records of spawning in hours after sunrise plotted against days after new year with monsoon period preventing surveys indicated in the middle of the year; B) Records of spawning in hours after sunrise against days from full moon.

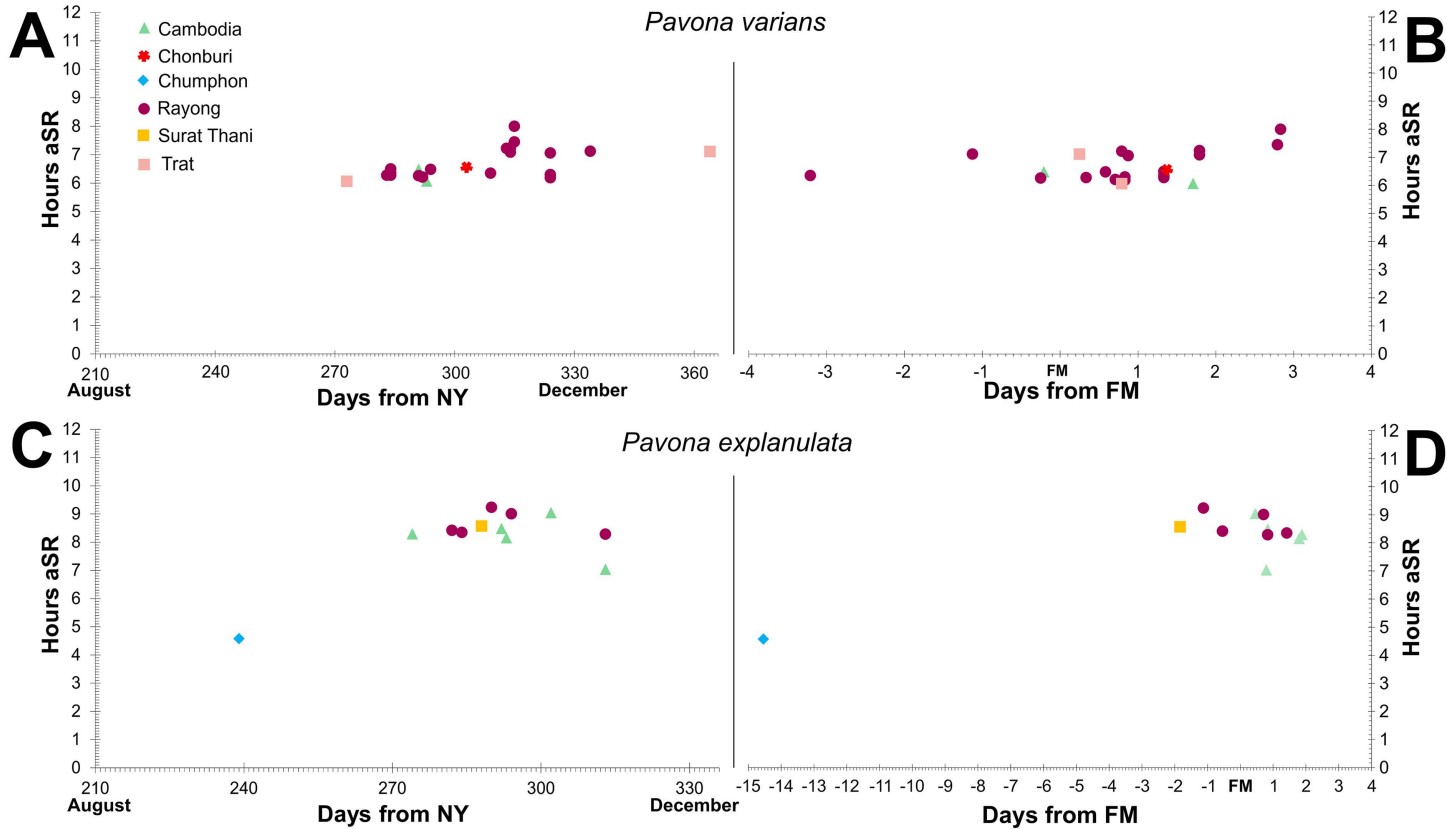

**Fig 4. Plots of spawning timing for *Pavona varians* and *P. explanulata* along the Gulf of Thailand (by province) and Cambodia between 2021 and 2024.** Spawning timing of *P. varians* with hours after sunrise plotted against (A) days after new year (emphasising the second half of the year) and (B) days after full moon, and *P. explanulata* showing hours after sunrise plotted against (C) days after new year and (D) days after full moon.

## Discussion

### Synchrony in the Gulf of Thailand

Based on spawning records from 14 genera from the Gulf of Thailand, our data provide strong credence for synchrony of coral spawning across the region, with compelling evidence to suggest spawning cues based on lunar and solar timing. The majority of spawning corals clustered tightly based on lunar phase (Fig 3B) and number of hours post sunrise (Figs 3,4). Our findings expand on earlier observations, revealing that *Pavona explanulata* and *P. varians* are daytime spawners across the Gulf of Thailand and are highly synchronised across the region. Null data and night-time surveys revealed that spawning of these corals is restricted (and thus predicted by) a narrow window of hours after sunrise and the timing of the full moon. The reproductive season for these corals across the Gulf of Thailand appears to largely peak in October and November, however there are indications of timing offset between the eastern and western Gulf. The coral spawning data presented here contributes to the limited prior documentation of synchronous mass spawning events from the Gulf of Thailand. Overall, our inferences on months of development based on observed reproductive output for night-time spawning corals agree with prior findings [38,46]. Additionally, the number of days after full moon for onset of coral spawning agree with records from Koh Tao ([43], Appendix A).

Questions remain regarding the close clustering of the timing of most *Pavona* spawning recorded from the Gulf of Thailand as distinct from those reported from seven colonies from Koh Rangka Jeaw in Chumphon province [30]. Based

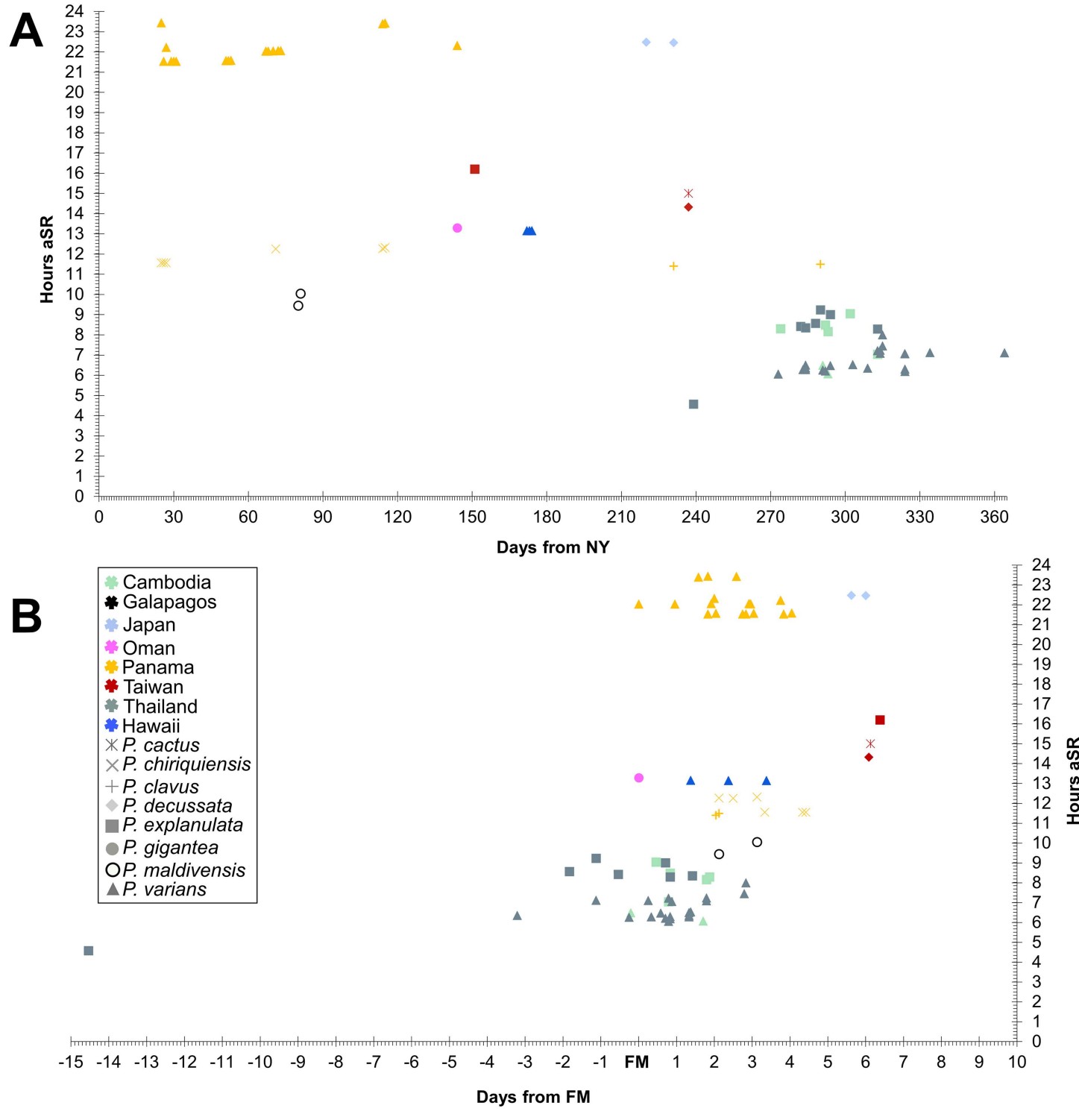

**Fig 5. Plots of spawning timing for *Pavona* corals across the globe by species and location.** A) Records of spawning in hours after sunrise plotted against days after new year; B) Records of spawning in hours after sunrise against days from full moon.

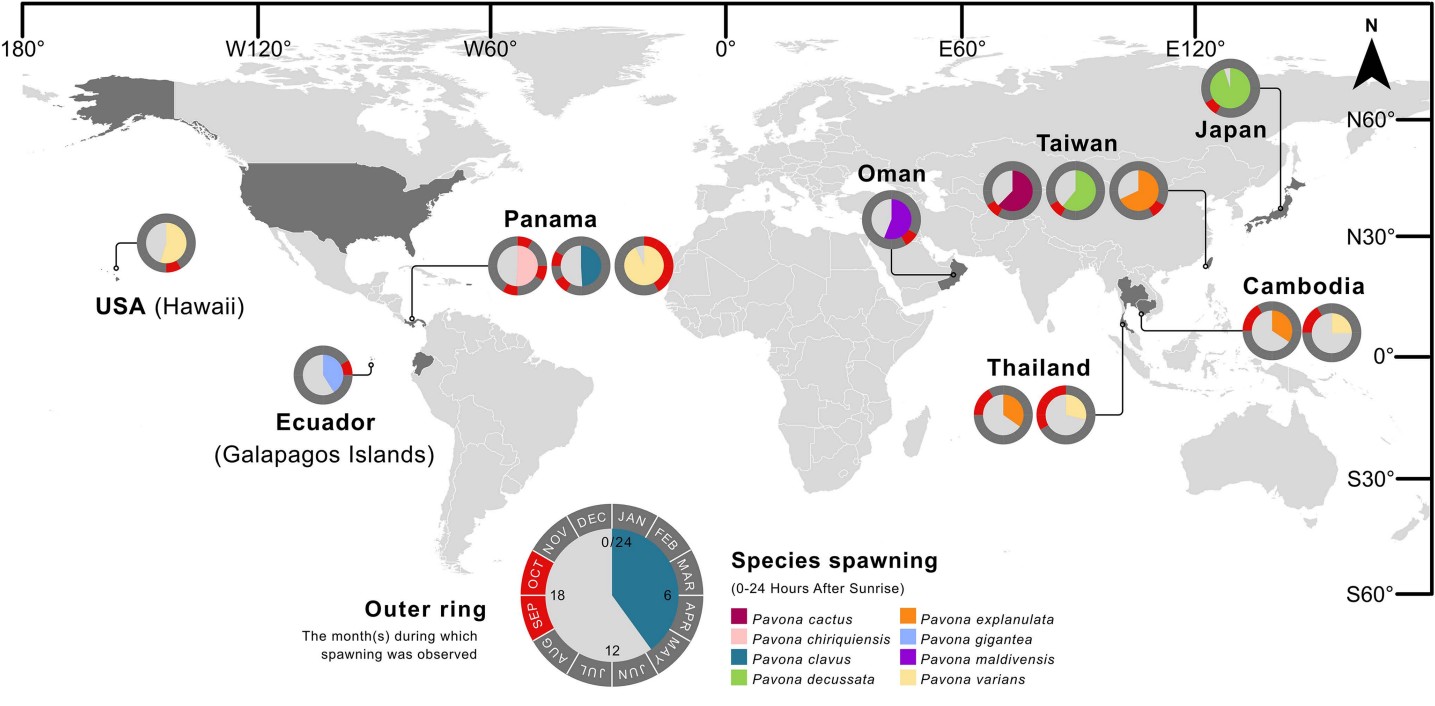

UPDATED MAP: 4th December 2025
Source: Base data from Natural Earth (public domain). Map modified and styled using QGIS 3.42 "Münster".
Natural Earth. Free vector and raster map data. Available at: https://www.naturalearthdata.com/ (public domain).

**Fig 6. Spawning timing of *Pavona* corals across the globe, by species.** Charts indicate briefest quantity of minutes and hours after sunrise (out of a total 24) at the start of spawning. Outer ring of the pie chart indicates the total months of the year during which spawning was identified per species per location. Legend provides an example of *Pavona clavus* indicating earliest start of spawning period to be approx. 9 hours and 50 minutes after sunrise, with all spawning records to be September and October. Detailed timings accessible in S1 Table. The basemap incorporates provincial boundary data from SimpleMaps [63,64] (CC BY 4.0).

on the single image shared from Chumphon [30] and our own surveys in the area, we maintain a high credence in our identification of earliest daytime spawning corals in Thailand corresponding to *P. explanulata*. Further exploration of *Pavona* spawning along this area are needed to explore possible differences in spawning time. Our dataset agrees with earlier observations [38,42,43] that night-time spawning periods differ for corals along the eastern and western Gulf of Thailand. Eastern Gulf night-time spawners typically spawn in the lunar cycles of February and March (with January and April being exceptionally rare), whereas western Gulf colonies typically spawn in March and April, with February and May records being almost absent. During days of concurrent observation, Rayong Province, Thailand and Koh Rong Sanloem Cambodia, onset of spawning was found to differ by 23 minutes for *P. varians* and 85 minutes for *P. explanulata*. Notably, on multiple occasions, where synchronicity was observed in spawning between Rayong and Trat provinces in Thailand, and Koh Rong Sanloem and Koh Koun in Cambodia, an absence of spawning was found at sites in Chonburi province. It is unclear if this is indicative of some localised disturbance of reproductive capacity, or if a breakdown in synchrony might be caused by other factors [67]. Lack of synchrony in spawning events has been shown to drastically reduce gene exchange between colonies [68–70] and therefore act as a driver for genetic divergence [70,71]. Divergences by as little as 1–3 hours in spawning time has been shown to lead to independent genetic clades in congeners [72]. The presently available data, while limited, supports synchrony in spawning within specific regions of the Gulf of Thailand, as well as to a lesser degree, between regions.

## Assessing non-invasive methods to study coral reproduction

Based on the data presented here, we have demonstrated the utility (and possibly the extent) of coral spawning data collection and prediction using in-situ observation-only techniques. While the sampling of colonies for the purpose of reproductive (i.e., gamete development) or taxonomic investigation will likely remain an essential tool, such activities should be carried out sparingly in areas where coral decline has been significant. Nonetheless, absence of spawning observations from other coral genera in our dataset highlights a limitation of the in-situ, non-invasive techniques employed in this study, suggesting that histological and gamete-development investigations remain the best tool for initial investigations, for taxa and regions where some baseline data is sparse or lacking. Our work thus far from Rayong and Surat Thani provinces ([43,50,68]; present study) allows for the identification of multiple abundant groups (e.g., *Astreopora* spp., *Leptastrea* spp., foliose *Pavona* spp., etc.) which comprise significant proportions of Thai coral communities [52] but for which regional spawning data is sparse or entirely absent. Indeed, our recent assessment of coral communities across much of Thai waters have identified *Pavona* (particularly foliose *Pavona* taxa) to be abundant throughout the region [52]. Given multiple years of surveys from across the Gulf of Thailand, the absence of spawning from these groups in the region (and sparsity from global data) highlight the value of dedicated histological assessments to investigate the role of timing among other factors on the reproduction of these corals.

## Global spawning patterns of *Pavona*

By far the most extensive assessment of reproductive dynamics of *Pavona varians* was carried out by Glynn et al. [36] along the Pacific coast of Panama. Based on an eight-year dataset of spawning observations and a long-term sampling regime investigating reproductive dynamics, *P. varians* was found to spawn shortly before sunrise (up to 2 hours before) and to be generally hermaphroditic with only a small portion of colonies being strictly gonochoric. However, the proportion of hermaphroditic vs. gonochoric colonies appeared to vary considerably across sites. The authors categorised the species as an 'alternating sequential hermaphrodite' Glynn et al. [36] where the spawning of one sex is preceded by the other within a given colony, lunar phase, and breeding season. These observations alongside additional data on gametogenic development also suggested nearly year-round reproductive activity across the range from Costa Rica, Panama and the Galapagos Islands. The earliest indications of daytime spawning in *P. varians* were recorded in a single-page abstract by Maté [32] where colonies from Hawaii were recorded to spawn prior to sunset at 19:05–20:15. In both these reports, the spawning of the species was recorded shortly before or after a change in illumination (sunrise or sunset) but thus far no observations have demonstrated spawning exclusively during periods of peak illumination (within a 24-hour cycle), as was recorded from the Gulf of Thailand. An investigation into gamete development of *P. varians* from the Red Sea inferred spawning to occur in April [73], further highlighting the distinction observed from the Gulf of Thailand. Additionally, the majority of *P. varians* spawning observations recorded across the Indo-Pacific occur between FM and 4 days after the lunar peak. The compiled dataset does appear to suggest that the onset of spawning within a given month is mediated by the lunar phase, however synchronicity across the calendar year and within a 24-hour period coincided with longitude separations more than latitude.It is possible that future observations during the survey months not covered in our work (May-August) may yield data that improves our understanding.

In contrast to *P. varians*, little is known about the reproductive biology of *P. explanulata*. Outside of the Gulf of Thailand, the only record of spawning we found in the literature for a coral identified as *Pavona explanulata* comes from Taiwan [61]. The authors provide a single record of night-time (9:30 pm) egg release six days after FM in May, however no further descriptions nor details are provided. Our observations differ considerably from this record in almost every way, except for colonies being identified as gonochoric. Our observations from the eastern Gulf of Thailand demonstrate a predictable onset of spawning shortly after *P. varians*, typically between 15:00–16:00, rarely extending beyond 17:00, and a definitive lack of spawning immediately prior to or after sunset (during surveyed periods). The earliest record of daytime spawning from the Gulf of Thailand [30], however, differs from the remainder of our dataset of observations for the species due to

spawning one day prior to the new moon, on 27th August 2003, during low tide. Curiously, the authors state this date to be the third day after the full moon, possibly suggesting an incorrect date of spawning, however for the purpose of this work we assume a misinterpretation of the lunar calendar. It is noteworthy that Glynn et al. [36] also identified a relationship between onset of spawning and the new moon as well as peak spring tides. While our study did not specifically investigate the relationship between tidal height and spawning time, the absence of spawning observations during new moon conditions across most of our dataset suggest that tidal influence in the Gulf of Thailand may be at most secondarily involved.

Given the wide range of spawning seasonality identified for *P. varians* across its range, it is unsurprising to find variability in spawning of *P. explanulata*. Indeed, when compared with other *Pavona* species, considerable variability is noted in the spawning period during the calendar year and within a given day. For example, alongside *P. varians*, Glynn et al. [36] also provide considerable data on *P. chiquiriensis* (initially identified as *Pavona* sp. prior to its description), which was found to spawn around sunset approximately 12 hours offset from *P. varians* from the same sites. Meanwhile *P. clavus* colonies from nearby Panamanian islands was also found to spawning around sunset [59], however consistently between August to October, as opposed to in January, April and July as recorded for *P. chiquiriensis.* Similarly, *P. cactus, P. decussata*, and *P. explanulata* have all been recorded to spawn after sunset in Taiwan, however the latter species spawning in May while the foliose taxa were found to spawn in August [58,61]. It is noteworthy that relatively little is known about the spawning of the abundant foliose members of the genus, with only four colonies of *P. cactus* and two colonies of *P. decussata* found in the literature, and nothing for the others. Our observations in this study lend credence to *P. frondifera* and *P. decussata* (abundant across the Gulf of Thailand) are possibly night-time spawning corals based on our null data. Across the genus-level, some possible trends of *Pavona* spawning are hinted at across the Indo-Pacific. Firstly, the vast majority of spawning observations appear to be in proximity to the lunar peak, with some possible indication that colonies spawning earlier in the day (hours after sunrise) are somewhat correlated with spawning earlier in the lunar cycle (Fig 5B). This of course is possibly skewed due to biases in observation time being higher between 3–7 days after the full moon. Across the biogeographic range (Fig 5A) there is a possible indication that, overall, colonies spawning later in the year are more likely to spawn earlier in the day (hours after sunrise). Of course, data remains lacking from the majority of locations from which *Pavona* corals are recorded and from many described species. Thus, the current dataset is far from a comprehensive review of the entirety of *Pavona* corals.

## Broader insights into daytime spawning

It is at present unclear if the consistent daytime spawning of *Pavona* corals in the Gulf of Thailand represents a recent population-wide adaptation or a localised shift in spawning time when compared to the remainder of spawning corals and locations. A key assumption made in the present study and review of global *Pavona* spawning, is the taxonomic stability of species identified by authors across the globe. Variation in reproductive outputs becomes considerably more likely if multiple cryptic species are represented within a single pool of data incorrectly attributed to a single species [36]. At present however, a comprehensive taxonomic review of the genus *Pavona* is somewhat limited by the need for considerably more data on ecological, morphological, and molecular data across much of its range [39]. The most recent species to be described in the genus, *P. giannii*, is yet another example of a taxon that bears close morphological resemblance to *P. varians*, likely leading to it being overlooked as an independent taxon to date [39]. With multiple junior synonyms of *P. varians* (*P. repens* (Brüggemann, 1877), *P. intermedia* (Gardiner, 1898), and *P. percarinata* (Ridley, 1883)) being described from different locations, future application of integrated morphological and molecular techniques to this group may challenge the currently accepted diversity of taxa.

The identity of *Pavona* spp. (S4 Fig) in our dataset was determined based on skeletal morphology, agreeing with the brief and un-illustrated original descriptions [74] and subsequent literature [75], based on in-situ examination of skeletons in recently killed polyps), ecology (species-specific associations as in [76,77]) and, in the case of *P. varians*, a nearly identical description of gametes released during spawning between our data and that of Glynn et al. [36]. Spawning during

the daytime raises significant questions regarding the adaptation of specific corals to changes in light availability in inducing spawning [21]. It is apparent from our dataset for example that the number of hours after sunrise is a more precise predictor for *P. explanulata* and *P. varians*, as opposed to hours before or after the full moon. This same aspect is notable in night-time spawning corals too which initiate spawning in general within a predictable window a given spawning night, regardless of month or year. There is no shortage of literature on importance of solar and lunar cycles in mediating night-time spawning time, and the varying roles of changes in light intensity and colour (wavelength) correlating with the onset of spawning timing [2,9,17,20,24–26,78,79]. If these mechanisms are to be considered as equally applicable to daytime spawning *Pavona* spp. in the Gulf, then this would imply a highly consistent 'countdown' several hours after dramatic change in illumination (sunrise, or perhaps sunset the previous day). Given the considerable variation of spawning time for *Pavona* spp. across the Indo-Pacific but high synchrony within populations, it is unlikely that this variation is a result of the major illumination-associated disruptor of synchrony, artificial light. Chronic exposure to artificial light, particularly white and blue light, have been shown to disturb gametogenic development in corals and disrupt synchrony [78,80–83]. The use of red-light is commonly applied to coral spawning monitoring in an effort to minimise disturbance of corals and other marine life, benefits of which are attributed to the shorter distance of attenuation of longer wavelengths [16,17,84]. In the present study, night-time SCUBA surveys were conducted using standard underwater torch light (white light) as opposed to using red-light filters, due to the already challenging prospect of spotting the near-transparent and potentially lethal individuals of *Chironex* sp. with white light, with red-light filters further reducing visibility and contrast considerably. Surveys using the ROV also used bright white light as surveys were restricted to in-built lighting with no modifications. While no studies have yet been conducted to assess the impact of short-term, close-proximity exposure to white light during spawning periods, the potential for a disruptive influence cannot be discarded. Nonetheless, patterns of spawning over the survey period appeared to be reinforced instead of being disrupted, and largely agree with earlier observations ([43], Appendix A).

Additionally, exposure of gametes to ultraviolet radiation of daylight spawners suggests possible photoprotective mechanisms that may differ considerably from conventionally night-time spawning corals. Little is known about the photoprotective mechanisms of gametes and larvae for agariciid corals, let alone differences between taxa based on spawning time or type, however insights may be gleaned from investigations into UV stress in non-agariciid taxa. Among the most informative may be the work of Aranda et al. [83], who reported differential UV tolerance in *Orbicella faveolata* (Ellis & Solander, 1786) larvae, with low sensitivity to UV radiation during the larval stage but high sensitivity in planulae. In contrast, Zhou et al. [85], found natural (overall) UV radiation had no noticeable effect on the survival, metamorphosis, or settlement on the larvae of *Seriatopora caliendrum* Hemprich & Ehrenberg, 1834. However, they did identify that that UV-A radiation alone did cause a decline in metamorphosis and settlement rates, which disappeared when UV-B was also included, which indicated a possible effect of UV-A on the duration of the planktonic phase of these larvae. Both these studies provide two examples of the seemingly widely differing and variable responses of coral larvae to different intensities and wavelengths of UV radiation among the few taxa that have been investigated [86–89], highlighting the need for more investigations on this topic.

In a recent investigation into the morphology of *P. varians* [90], the authors identified the species as one of a minority of taxa that may benefit from deeper water thermal refugia due to its light associated plasticity. It is possible therefore that this same plasticity and adaptation may extend to gametes and juveniles, allowing for a wide range of spawning timings. A possible explanation emerges, therefore, for the local adaptation to earlier daytime spawning in the Gulf of Thailand. Our surveys consistently found both *P. varians* and *P. explanulata* predominantly at greater depths of the reef slope, often as a 'band' of colonies along the edge of fringing reefs. Colonies were only found in shallower water when shaded by overhanging corals or rocks, or at high-turbidity sites. The Gulf of Thailand is well known to be a high-turbidity region due to the multiple major rivers exiting into the Gulf [91–94]. This may therefore drive localised 'availability' of daylight hours allowing for a broader window of spawning for these species, however, this is not seen in other high-turbidity regions, nor

would it explain the noon-time spawning observed in our data (a period of peak illumination). This theory is also unsupported by the spawning patterns of the majority of other coral taxa in our dataset, which show minimal diverge from the spawning timings found elsewhere in the West-Pacific [33]. An alternative explanation may lie in the possible trade-off between illumination and thermal load, with *Pavona* corals spawning far later in the year than all other groups in the Gulf, and congeners across the globe. Water temperatures in the Gulf are typically coolest between October and January, thus spawning during the cooler months may reduce direct thermal-stress to gametes and larvae. Recent documentation from Ningaloo Reef, Western Australia [95] identified eutrophication and subsequent deoxygenation leading to widespread fish mortality associated with elevated temperature post-spawning and gametes/larvae being unable to disperse. Cooler temperatures may thus provide an ecological benefit to the survival of larvae in addition to a thermal benefit. However, this again does not explain the prevalence of daytime spawning during the cooler months, with night-time conditions obviously receiving no direct solar irradiation.

While daytime spawning of scleractinian corals is far from the norm, numerous (and growing) examples of sporadic or periodic, isolated or synchronised, daytime spawning cases are documented in the literature, often however as brief descriptions [27–30,44,96–98]. Among these are cases of Fungiidae corals releasing gametes ranging from a few hours after sunrise [29] to shortly before and during sunset [43,98], and widespread synchronised daytime spawning in *Porites rus* [21,27]. Our documentation represents the first time *Cycloseris* corals have been found to spawn in Thailand, and thus it is unknown how widespread daytime spawning may be for the genus. For *Porites* however, a closer examination is needed to ascertain the species identity of colonies at Sail Rock spawning during the daytime to investigate if asynchrony is due to taxonomic or some other localised variation. Given the remoteness of the single site from where it was recorded, Sail Rock, 15 km from the nearest coastal illumination, it is unlikely to be affected from coastal development-induced artificial lighting. However, abundant night-time squid fishing activities are carried out offshore the islands of the Samui archipelago (and the wider Gulf of Thailand) which utilise extremely bright lights visible from kilometres away. The possible impact of night-time illumination used in such fishing activities on coral spawning general reef ecology needs further study, and may disproportionately affect isolated pinnacle sites where offshore fishing is permitted.

Further investigation is needed for the geographical extent of daytime *Pavona* spawning. Questions also arise whether fertilisation and recruitment remain viable under UV exposure or if the daytime spawning nature of the population interrupts the population turnover of these corals. Presence and even growth in reef cover is not indicative of population turnover or recruitment [99], and thus abundances of mature colonies, recruits and larvae should be assessed independently for species-level population assessments. Glynn et al. [36] identified recruitment of *Pavona* corals to be limited by unusually high temperature anomalies, at thresholds of between 1.6 and 1.9 C°, but is unexpectedly recruitment was favoured by anomalies of 0.5 to 1.5 C°. In the case of the Gulf of Thailand, the spawning period of *P. varians* and *P. explanulata* being offset by at least five months has major implications for the dispersal of larvae, due to significant hydrodynamic changes at different times of the year [100–102]. Our study did not document fertilisation and dispersal kinetics, however *P. varians* larvae have been known to survive at least up to 10 days [36] allowing for a significant window of dispersal between source and sink sites. Indeed, little is known about the status or biogeography of Agariciid taxa across Thailand, with population changes being typically documented at highly localised scales and rarely with species-specific differentiation [103,104]. In a recent assessment on the efficacy of micro-fragmentation techniques for coral restoration [105], *P. varians* was found to have the slowest growth rate of the four taxa and significantly higher mortality than *Diploastrea heliopora, Lithophyllon undulatum* (but near-identical to *P. decussata*). The findings of regular synchronised spawning of *P. varians* documented in the present work provide a baseline for possible larval propagation or similar sexual-reproduction conservation efforts to be carried out in the Gulf of Thailand. Such techniques are increasingly being utilised [106,107] as an alternative to the widespread and often damaging (when unregulated) asexual fragmentation and propagation techniques, often freely used in Thailand and the wider tropics which dilute local genetic diversity and likely reduce fertilisation rates due to saturation of reef areas with fewer genotypes [108,109].

 

At present, coral spawning observations in the Gulf of Thailand rely on incidental observations or predictions from previous years, and do not provide the precision allowed by direct assessment of internal gametogenic development by sampling. Therefore, spawning timing and predictive potential for many genera remain unknown and may require more active examination. Alternatively, temporary relocation of medium sized colonies to ex-situ conditions may allow for assessment without the restrictions of microscopy, SCUBA, or safety concerns from envenomation. In a recent survey [66] utilising environmental DNA (eDNA), daytime spawning corals were reliably recorded in daytime water samples but not in nighttime samples. Such an eDNA approach may therefore be a valuable tool in identifying the diversity and seasonality of day and night-time spawning corals across a wider geographic range.

## Supporting information

**S1 Table. Coral spawning records observed along the Eastern Gulf of Thailand during the five-year survey period (2020–2024), compiled alongside published records of prior spawning in the Gulf of Thailand and *Pavona* spawning from across the Indo-Pacific.**
(XLSX)

**S2 Fig. Setting of gamete bundles in *Acropora humilis* (A), release of gamete bundles in *Platygyra sinensis* (B), an individual of *Chironex* sp. observed during coral spawning surveys (C), the Fifish V6 ROV used for coral spawning surveys (D), daytime sperm release of *Porites* (taken by Evgeny Kovban) at 10:53AM on 08th April 2023 (E,F).**
(JPEG)

**S3 Fig. Full frame screenshots taken from ROV footage during coral spawning surveys.** Gamete bundle release of *Goniastrea*, example of released bundles shown by arrows (A), sperm release by *Porites* (B), setting of gamete bundles prior to spawning in *Platygyra* (C) and *Favites* (D). Scale bars not provided due to lack of scaling tools associated with ROV.
(JPEG)

**S4 Fig. Colonies of *Pavona varians* (A, scale bar 30 cm; C-E, scale bar 10 mm) and *P. explanulata* (B, F, scale bar 10 mm).**
(JPG)

## Acknowledgments

We would like to thank Trent McGrath, Thomas Thana Real, Claudia Marcellucci, Maythira Kasemsant, Adam Stoddard, Tim McCabe, Stephanie Rog, Manua Chazerand, Tharamony Ngoun, Chantha Chroeng, Srenh Sorn, and all the survey staff for their assistance in surveys in this project. This project was made possible thanks to logistical support Thai Ocean Academy Koh Chang, Thai Ocean Academy Bangkok, the Royal Thai Department of Marine and Coastal Resources. We are grateful to the Coral Spawning Database team including Alasdair Edwards, James Guest, and Andrew Baird for their input at the early stages of the project. Finally, this work is in loving memory of Andrea Heather Go, who played an essential role in all aspects of this project, and succeeded in becoming a passionate and capable marine biologist.

## Author contributions

**Conceptualization:** Rahul Mehrotra.

**Data curation:** Rahul Mehrotra, Suppakarn Jandang, Coline Monchanin, Natchanon Kiatkajornphan, Supatcha Japakang.

**Formal analysis:** Rahul Mehrotra, Suppakarn Jandang.

**Funding acquisition:** Rahul Mehrotra, Suppakarn Jandang.

**Investigation:** Rahul Mehrotra, Coline Monchanin, Matthias Desmolles, Lalita Putchim, Natchanon Kiatkajornphan, Supatcha Japakang, Anne Groenevelde, Morokot Long, Matthew Glue, Nitchanan Nilkerd.

**Methodology:** Rahul Mehrotra, Suppakarn Jandang.

**Project administration:** Rahul Mehrotra, Suppakarn Jandang.

**Resources:** Rahul Mehrotra, Suppakarn Jandang.

**Supervision:** Rahul Mehrotra.

**Validation:** Rahul Mehrotra, Suppakarn Jandang.

**Visualization:** Rahul Mehrotra, Suppakarn Jandang, Supatcha Japakang, Anne Groenevelde.

**Writing – original draft:** Rahul Mehrotra, Suppakarn Jandang, Coline Monchanin, Matthias Desmolles, Lalita Putchim, Natchanon Kiatkajornphan, Supatcha Japakang, Anne Groenevelde, Morokot Long, Matthew Glue, Anchalee Chankong, Nitchanan Nilkerd, Laddawan Sangsawang, Vincent Pardieu.

**Writing – review & editing:** Rahul Mehrotra, Suppakarn Jandang, Coline Monchanin, Matthias Desmolles, Lalita Putchim, Natchanon Kiatkajornphan, Supatcha Japakang, Anne Groenevelde, Morokot Long, Matthew Glue, Anchalee Chankong, Nitchanan Nilkerd, Laddawan Sangsawang, Vincent Pardieu.

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
