## [Decision Letter · Decision Letter 0]

5 Nov 2025

Dear Dr. Jandang,

Thank you for submitting your manuscript to PLOS ONE. After careful consideration, we feel that it has merit but does not fully meet PLOS ONE’s publication criteria as it currently stands. Therefore, we invite you to submit a revised version of the manuscript that addresses the points raised during the review process.

We look forward to receiving your revised manuscript.

Kind regards,

Parviz Tavakoli-Kolour

Academic Editor

PLOS ONE

Journal Requirements:

“This research was supported by Love Wildlife Foundation, Thailand and Fauna and Flora International, Cambodia.”

3. We note that Figure 1 and 6 in your submission contain map images which may be copyrighted. All PLOS content is published under the Creative Commons Attribution License (CC BY 4.0), which means that the manuscript, images, and Supporting Information files will be freely available online, and any third party is permitted to access, download, copy, distribute, and use these materials in any way, even commercially, with proper attribution. For these reasons, we cannot publish previously copyrighted maps or satellite images created using proprietary data, such as Google software (Google Maps, Street View, and Earth). For more information, see our copyright guidelines: http://journals.plos.org/plosone/s/licenses-and-copyright.

1. You may seek permission from the original copyright holder of Figure 1 and 6 to publish the content specifically under the CC BY 4.0 license.

Reviewers' comments:

Reviewer's Responses to Questions

**Comments to the Author**

1. Is the manuscript technically sound, and do the data support the conclusions?

Reviewer #1: Partly

Reviewer #2: Yes

Reviewer #3: Yes

2. Has the statistical analysis been performed appropriately and rigorously?

Reviewer #1: N/A

Reviewer #2: N/A

Reviewer #3: N/A

3. Have the authors made all data underlying the findings in their manuscript fully available?

Reviewer #1: Yes

Reviewer #2: Yes

Reviewer #3: Yes

4. Is the manuscript presented in an intelligible fashion and written in standard English?

Reviewer #1: Yes

Reviewer #2: No

Reviewer #3: Yes

Reviewer #1: The paper presents relevant information on coral spawning in an understudied location, and is generally well-written. Although I recognize the significance of this research, I have one major issue with the paper and this is about the inclusion of null data, specifically with the taxon (see Table S1). It is crucial to identify the spawning organism, and entering null data renders it untrustworthy. I would advise excluding these data unless the authors were able to identify them, even at the lower taxonomic level (e.g. family). Additionally, there are a lot of null data for the taxon (396, more than 50% of the listed taxon), which raises doubts about whether the observations are actually from hard corals or from other taxa. I have chosen to reject this paper for the time being because this information is the foundation of this study. I do, however, advise the authors to submit again after the identification of the species have been resolved. I also added some more suggestions/clarifications below.

Title: I feel that the title is misleading. You also discussed about other spawning coral species and night time spawning in the gulf and not only Pavona, therefore, I suggest revising this title appropriately.

Abstract

Line 30: Looking at your data (Table S1 and Fig. 4), it does not seem to be synchronous spawning across the gulf but to certain locations and to some colonies. For example, Pavona explanulata spawned on June 15-16 and 18-19, 2024 at Koh Koun, on June 16, 2024 at Hin Yuan, on June 17, 2024 at Koh Rong Sanloem. In 2022, it spawned on October 9 at Hin Yuan, October 11 at Hin Ploeng, November 9 at Hin Yuan and Koh Koun. Please revise accordingly.

Introduction

Line 89: Please cite these few studies here.

Line 97-100 I would use caution while making this claim and advise the authors to reword it. Removing an entire colony from the reef is not less destructive.

Lines 90-101: Rather than dedicating this whole paragraph by explaining the various methods of coral reproduction, I suggest that this should be rewritten focusing on mass coral spawning in Thailand. In fact, while reading Line 89, I have expected that the next sentences would be about mass spawning or coral reproduction in the Gulf of Thailand.

Results

Line 201-202: Did all spawning occur in all years of observation? If not, please add the years here when spawning was observed.

Line 248: Suggesting that these are brooding over broadcast spawning is an overspeculation and should not be written in results. Perhaps you can mention this in the discussion with an added rationale. Also, sperm spawning may suggest gonochoric sexuality but not reproductive mode.

Line 259-260: This is discussion material and should be removed from here.

Table S1: I highly recommend that null data for the taxon level should be replaced with species identity even in the lowest possible resolution, for instance family.

=Suggest revising Notes to “Notes or Reference/s”

=Be consistent with the formatting of your dates.

=Please italicize species name. Also check species name where the specific epithet has been capitalized (e.g. Pavona Explanulata).

= I would recommend restructuring this table for better organization, perhaps arranged by species, or date of observations.

= Should be “Table S1. Coral spawning records observed along the Eastern Gulf of Thailand during the five-year survey period (2020-2024), compiled alongside published records of Pavona spawning from across the Indo-Pacific”

Figure 2: Do you have a better photo of c? Current photo is blurry or perhaps its just my copy.

Figure 3: I think it’s inappropriate to include night time spawners in this figure if you aim to use sunrise as reference of spawning. Perhaps divide figures for daytime and night time spawners (this using hours after sunset).

Figure 5: Same comments with Fig. 3

Discussion

Line 310: Be careful with the use of synchronous mass spawning. Your results did not show synchronous spawning if you look closely. Spawning 784 colonies and previous studies do not occur at a brief period of time, but across months and years. Indeed, you documented various spawning events with a number of colonies and species spawning at around the same time. But the overall pattern did not show synchrony.

Line 311: You did not determine development and measure reproductive output in this study so I recommend removing this part.

Line 318-320: I strongly believe you should revise this. You did not conduct spawning observations prior to September, so the term lack of data is more appropriate than considering this previous observation as outlier.

Line 334: You mentioned Agaraciidae and non- Agaraciidae multiple times in your manuscript, thus for better readability, perhaps it’s better to add another column in your Table S1 for coral family.

Line 345-356: This is more like a conclusion/recommendation piece, thus I would suggest placing or incorporating this in the paragraph at the end of your discussion.

Line 357: I suggest adding Pavona in your subheading. Additionally, some portions of this part appeared to be too long or with some irrelevant information (e.g., Lines 363-367), related to your study

Line 365: Please add reference/s on this.

Line 374-375: It should be worth mentioning that spawning observations was not conducted in May to August as opposed to other mentioned studies.

Line 417: correlated.

Line 440: Moon phases (i.e. full moon) is a recognized cue for the day/night of spawning but not on exact time (hour) of the day. Maybe perhaps you can discuss on other cues of exact time such as temperature, tidal cycles, photoperiod, etc.

Line 497: needed to ascertain

Reviewer #2: The manuscript "Daytime spawning corals in the Gulf of Thailand and a review of spawning patterns

in Pavona corals across the Indo-Pacific" offers invaluable information on scleractinian coral spawning in the Gulf of Thailand over multiple years, with specific focus on daytime spawning Pavona species. It sheds light on taxa that are often overlooked because they do not spawn during the conventional spawning time that is after sunset. By paying attention to the daytime spawners and employing creative solutions, the authors demonstrate that the overall coral spawning periods in the Gulf are between September to December and February to April. Curation of null data also allowed for further refinement of the spawning periods in this target region. The data comparison between other regions highlights that regional differences exist for month of spawning and time of the day, but not for days relative to full moon. These findings will greatly benefit researchers in coral reproductive biology and restoration projects, and advance the overall knowledge in coral spawning. In the attached document, I list questions and minor suggestions that may help improve the manuscript.

Reviewer #3: You document clear, daytime, synchronous spawning windows for Pavona explanulata and P. varians in the eastern Gulf of Thailand/Cambodia and add a useful global recap for Pavona. The findings are relevant and actionable. Most of my notes are about how the figures and wording could be tighter.

1. Fig. 1&6. Right now the plot shades from 0 h (sunrise) all the way to the “latest onset,” which reads like “spawning might start anytime from 0–X h.” If the real window is 6–8 h after sunrise, only shade 6–8 h. Add a tick for the median and a band for IQR/95% CI if it is available.

2. Line 49-51. The current sentence implies both modes disperse “gametes” (eggs + sperm) by currents. That fits broadcast spawners but not brooders (who retain eggs, release sperm, fertilize internally, then release larvae).

3. Most Dipsastraea are hermaphroditic broadcast spawners with nocturnal bundle release. Daytime sperm-only is atypical and can be confounded by upstream sperm plumes or viewing angle.

Suggestion: label as “Sperm (putative)”, describe your scoring criteria (continuous release from the focal colony, no bundles observed, clear source), and discuss it conservatively.

**Do you want your identity to be public for this peer review?** For information about this choice, including consent withdrawal, please see our Privacy Policy

Reviewer #1: No

Reviewer #2: No

Reviewer #3: No

---

## [Author Response · Author response to Decision Letter 1]

25 Jan 2026

Reviewer #1: Please note that we have attached several new figures with clearer images. These can be found in the attached figures included in the response-to-reviewers files.

The paper presents relevant information on coral spawning in an understudied location, and is generally well-written. Although I recognize the significance of this research, I have one major issue with the paper and this is about the inclusion of null data, specifically with the taxon (see Table S1). It is crucial to identify the spawning organism, and entering null data renders it untrustworthy. I would advise excluding these data unless the authors were able to identify them, even at the lower taxonomic level (e.g. family). Additionally, there are a lot of null data for the taxon (396, more than 50% of the listed taxon), which raises doubts about whether the observations are actually from hard corals or from other taxa. I have chosen to reject this paper for the time being because this information is the foundation of this study. I do, however, advise the authors to submit again after the identification of the species have been resolved. I also added some more suggestions/clarifications below. Title: I feel that the title is misleading. You also discussed about other spawning coral species and night time spawning in the gulf and not only Pavona, therefore, I suggest revising this title appropriately.

-Authors Response: We greatly appreciate the reviewer’s efforts to improve the quality of our work and the detailed comments made. We concede the reviewer’s point that this paper does indeed talk about other spawning corals. We note however:

o There is a clear and distinct emphasis on Pavona corals on the majority of figures, tables, and results of our work.

o Other spawning corals are included for context, and partially draw on previously published literature.

Nonetheless, we have reworded the title to be more inclusive, while retaining the clear emphasis of our work. We have responded to the reviewer’s questions and confusion regarding the methodology, including use of null data, among other aspects in the responses below. The lack of null data is a well-known issue in identifying spawning periods, and we acknowledge that the null data cannot confirm absence and presence of every single coral taxa evenly per survey. What it absolute does do is narrow windows of possible spawning periods, as incidental observations of spawning are a) precisely how initial patterns for this project were found and b) reported with sufficient frequency and reliability to be shared as valid novel findings. We are optimistic that the reviewer’s concerns are adequately addressed below.

Abstract Line 30: Looking at your data (Table S1 and Fig. 4), it does not seem to be synchronous spawning across the gulf but to certain locations and to some colonies. For example, Pavona explanulata spawned on June 15-16 and 18-19, 2024 at Koh Koun, on June 16, 2024 at Hin Yuan, on June 17, 2024 at Koh Rong Sanloem. In 2022, it spawned on October 9 at Hin Yuan, October 11 at Hin Ploeng, November 9 at Hin Yuan and Koh Koun. Please revise accordingly.

-Authors Response: We appreciate the reviewer’s concern regarding our assessment of synchrony, however it appears the reviewer has perhaps misread the aforementioned figure and table. Neither the table nor the figure show any spawning in June (which could not have been recorded anyway as this was during the inaccessible monsoon period, as mentioned in the methodology). To improve clarification, we have updated table S1 to avoid confusion regarding the date format across the table, however these may easily be cross-referenced with the original table accessible to the reviewer for any further clarification needed. Regarding Figure 4, we remind the reviewer that the x axis begins at the month of August and thus cannot show data from June. We have, however, updated our caption for Fig. 4 to improve clarity on this point and we appreciate the reviewer demonstrating the need for this. Finally, the majority of our paper discusses and references the role of lunar, solar, and other environmental variables on the synchrony of coral spawning. Regarding the reviewer’s concerns regarding the offset of days and months in the synchrony of spawning in our data, we note that, in many (but we concede not all) cases where surveys were indeed carried out concurrently, synchrony of spawning was absolutely observed across the provinces and between the countries. This re-emphasises the value of the null data (where available) such that readers and reviewers may see where surveys were carried out but spawning was explicitly not seen, versus an absence of data due to lack of a survey. Secondly, synchrony within the coral spawning literature does not only apply to within a window of minutes/hours, but spawning events may also be considered synchronous if they spawn across the same spawning period (multiple colonies, per night, within the hours/days preceding or following the full moon or new moon). The offset across multiple months is entirely normal within coral spawning literature, and does not indicate a lack of synchrony, just that spawning events may be (and typically do) spread across more than one lunar cycle (most months typically only having a single lunar peak and so on). A lack of synchronisation is noted when the same environmental and biological cues no longer apply to observed spawning, which we also make some effort to discuss in our work.

Introduction Line 89: Please cite these few studies here.

-Authors Response: We have moved the sentences that cite this work to the relevant section of the paragraph and have divided the remainder to improve the follow of the introduction for the reader.

Line 97-100 I would use caution while making this claim and advise the authors to reword it. Removing an entire colony from the reef is not less destructive.

-Authors Response: The reviewer’s opinion that a temporary removal and return of a coral colony from the reef is not less destructive than permanent removal of often large sections of a colony, is an entirely fair one. While the authors’ disagree with this opinion, it does not change that fact that the second word used in the sentence is ‘’potentially’’, to avoid framing in absolute terms, and is based upon communications with local practitioners who do consider the practice ‘’less destructive’’. We also do not necessarily agree with the opinions of these practitioners, instead framing the practices as distinct and having it’s own nuances. In this section, we have not established the details nor generalised the a) size of colonies, b) size of fragments, c) short-term vs. long-term costs vs. benefits, etc. The purpose of this paragraph is to present context and concepts for the practices that have been and often are being conducted, that are relevant to our survey methodology as well as our study sites as a whole. Given that we have not only cited references, but also used an abundance of caution in our framing, (e.g. the following sentence explicitly raises concerns on the potential damage of this practice, aligning more with the reviewer’s perspective), we have not removed nor reworded this section).

Lines 90-101: Rather than dedicating this whole paragraph by explaining the various methods of coral reproduction, I suggest that this should be rewritten focusing on mass coral spawning in Thailand. In fact, while reading Line 89, I have expected that the next sentences would be about mass spawning or coral reproduction in the Gulf of Thailand.

-Authors Response: We have worked to improve the readability of this section and moved the concluding sentences and citations that specifically talk about spawning observations to the follow the start of this paragraph, with a natural lead into the gamete-development assessments and the relevant survey methods. We note however that this paragraph does not talk about ‘’various methods of coral reproduction’’ but rather the methods used by surveyors to assess this and predict coral spawning timing. We hope the clearer distinction between the earlier spawning records (the entirety of which have been stated in the appropriate line) and the techniques used to determine reproductive period.

Results Line 201-202: Did all spawning occur in all years of observation? If not, please add the years here when spawning was observed.

-Authors Response: Indeed, spawning was observed in all surveyed years. We have amended this section to clarify this for readers.

Line 248: Suggesting that these are brooding over broadcast spawning is an overspeculation and should not be written in results. Perhaps you can mention this in the discussion with an added rationale. Also, sperm spawning may suggest gonochoric sexuality but not reproductive mode.

- Authors Response: We appreciate the reviewer’s suggestion on this point, and while the entirety of this part of the results explicitly explores the differences between gonochoric and hermaphroditic sexuality, the hypothesising of reproductive mode is best explored in the Discussion if needed. We have thus removed the wording “possibly indicating brooding over broadcasting”.

Line 259-260: This is discussion material and should be removed from here.

Authors Response: We appreciate the reviewer’s suggestion, we have removed the indicated sentence from the Results section.

Table S1: I highly recommend that null data for the taxon level should be replaced with species identity even in the lowest possible resolution, for instance family.

Authors Response: Unfortunately, this is not possible and would reduce the value of our work, due to the methodology employed (clarified above). The surveys carried out across the reefs, including our night-time surveys and all surveys where spawning *was* documented, incorporated all corals and other organisms in the area where possible. Null data there explicitly indicates an absence of any spawning from any observed corals. While we can appreciate that this does not easily allow the reviewer to note which corals were not found to be spawning, we have amended our manuscript with multiple references to our other published material that more directly and comprehensively cover the coral community composition of the sites surveyed. A repeat of this data would not be able to fit in the supplementary table, and would largely amount to a direct duplication of the same data shared from earlier work. Instead, we have opted to *discuss* the absence/presence of spawning corals, in direct reference to this.

=Suggest revising Notes to “Notes or Reference/s”

- Authors Response: Thank you for this suggestion, we have made this edit.

=Be consistent with the formatting of your dates.

- Authors Response: Thank you, we have reviewed the table and made this amendment throughout.

=Please italicize species name. Also check species name where the specific epithet has been capitalized (e.g. Pavona Explanulata).

-Authors Response: Thank you, we have reviewed the table and made this amendment throughout.

= I would recommend restructuring this table for better organization, perhaps arranged by species, or date of observations.

- Authors Response: We appreciate the reviewer’s recommendation and have made numerous edits to allow the data to be filtered in numerous ways. However, the overall order of observations remains as presented (with each record having a specific number) such that cataloguing and review as per our data organisation is maintained. We trust that readers will have the capacity to use the in-built data filtration system to prioritise organisation based on specific needs.

= Should be “Table S1. Coral spawning records observed along the Eastern Gulf of Thailand during the five-year survey period (2020-2024), compiled alongside published records of Pavona spawning from across the Indo-Pacific”

- Authors Response: We have amended this based on the reviewer’s suggestion, with the addition of ‘’ …compiled alongside published records of prior spawning in the Gulf of Thailand and Pavona spawning from across the Indo-Pacific”

Figure 2: Do you have a better photo of c? Current photo is blurry or perhaps its just my copy.

-Authors Response: Unfortunately, we are faced with two challenges here in assisting the reviewer’s entirely valid concern. Firstly, it is quite likely that the Figure images made available to reviewers have undergone the usual heavy compression from submission of raw image files. Therefore, in an effort to allow the reviewer to properly

assess the resolution of our figures, we have included them, with captions, at the end of our responses to reviewers. We have also copied a snapshot of the relevant part of our updated Figure 2 in this response to more easily allow for further review. Secondly, the egg release in P. explanulata in our observations was highly challenging to capture clearly due to the size of the gametes, form of release, as well as the lack of distinctive pigmentation allowing for sufficient contrast. To aid in visualisation, we have amended this figure (and caption) to more clearly indicate the release of eggs (Please see the attached figures in the response-to-reviewers files).

Figure 3: I think it’s inappropriate to include night time spawners in this figure if you aim to use sunrise as reference of spawning. Perhaps divide figures for daytime and night time spawners (this using hours after sunset). Figure 5: Same comments with Fig. 3

-Authors Response (Fig. 3 and 5): We appreciate the reviewer’s concern that the hours after sunrise may not be applicable for night time spawning. However, the reviewer’s assumption that sunrise timing does not influence nighttime spawning is unfounded and unsupported by the literature (as discussed in our paper). Hours after sunset has historically been the framing for spawning timing, however there is no evidence to suggest that this in any way adds more accuracy or relevance than sunrise. We find that complicating visualization into multiple figures purely to follow historical convention does not add value. Adherence to a historical lens to view coral spawning adds a

historic bias, and we instead here find value in re-visualising all spawning through a different lens without changing the data itself. It is no less convenient than using sunset time, and indeed our data suggests that sunrise time may indeed be a more appropriate, if not more precise, cue to the onset of spawning. Of course, this particular hypothesis needs considerably more work to validate, but is nonetheless a valid framing of our data. Therefore, in lieu of greater justification than subjective convenience to the reviewer, we have left these axes and figures unchanged beyond font size and readability.

Discussion Line 310: Be careful with the use of synchronous mass spawning. Your results did not show synchronous spawning if you look closely. Spawning 784 colonies and previous studies do not occur at a brief period of time, but across months and years. Indeed, you documented various spawning events with a number of colonies and species spawning at around the same time. But the overall pattern did not show synchrony.

- Authors Response: As per our earlier response, this comment is unfortunately reflected by the reviewer’s misunderstanding of the data and visualisations provided (see response re: Table S1 and Fig. 4 above). As a result, we have attempted to make numerous modifications throughout this work to more easily allow readers to navigate the scale and nuances of the data provided. Again, we believe we have demonstrated with a high credence that multiple colonies spawning within minutes to hours of each other, within a single site, across sites, and seemingly across nations, over multiple years, and seemingly at no other times (though we have already eluded to margins of error) does leave room for little doubt that synchrony is absolutely present. Given this point being a somewhat subjective disagreement of opinion and interpretation based purely on data, and our conclusion (and central premise) of mass synchronisation, we have not made changes to this section.

Line 311: You did not determine development and measure reproductive output in this study so I recommend removing this

---

## [Editor Report · Decision Letter 1]

4 Feb 2026

Coral spawning patterns in the Gulf of Thailand reveal synchronised annual daytime spawning, with a review of spawning patterns in Pavona corals across the Indo-Pacific

PONE-D-25-47186R1

Dear Dr. Jandang,

We’re pleased to inform you that your manuscript has been judged scientifically suitable for publication and will be formally accepted for publication once it meets all outstanding technical requirements.

Kind regards,

Parviz Tavakoli-Kolour

Academic Editor

PLOS One
---

## [Editor Report · Acceptance letter]

PONE-D-25-47186R1

PLOS One

Dear Dr. Jandang,

I'm pleased to inform you that your manuscript has been deemed suitable for publication in PLOS One. Congratulations! Your manuscript is now being handed over to our production team.

Kind regards,

on behalf of

Dr. Parviz Tavakoli-Kolour

Academic Editor

PLOS One